# Discovering Generalizable Spatial Goal Representations via Graph-based Active Reward Learning

**Aviv Netanyahu**[∗]**, Tianmin Shu**[∗]**, Joshua B. Tenenbaum, Pulkit Agrawal**
Massachusetts Institute of Technology
Cambridge, MA 02139, USA
{avivn,tshu,jbt,pulkitag}@mit.edu

## Abstract

In this work, we consider one-shot imitation learning for object rearrangement tasks, where an AI agent needs to watch a single expert demonstration and learn to perform the same task in different environments. To achieve a strong generalization, the AI agent must infer the spatial goal specification for the task. However, there can be multiple goal specifications that fit the given demonstration. To address this, we propose a reward learning approach, Graph-based Equivalence Mappings (GEM), that can discover spatial goal representations that are aligned with the intended goal specification, enabling successful generalization in unseen environments. We conducted experiments with simulated oracles and with human subjects. The results show that GEM can drastically improve the generalizability of the learned goal representations over strong baselines.

## 1 Introduction

To build AI agents that can assist humans in real world settings, we have to first enable them to learn to perform any new tasks defined by a human user. To achieve this, an AI agent has to acquire two types of key abilities: i) the ability to develop a generalizable understanding of the goal or task specification intended by the human user and ii) the ability to plan or learn a policy for a given goal. In this work, we aim at engineering the first key ability for an AI agent. In particular, we focus on object rearrangement tasks, a type of common tasks studied in robot planning, where an agent must reason about the spatial goals that define a set of desired spatial relationships between objects. For instance, to set up a dinner table, one has to know how to place the plates and utensils appropriately.

One way to train agents to perform a new object rearrangement task is to provide manual goal specification, such as detailed instructions or hand-craft reward functions. However, creating a manual definition for the goal requires expert knowledge, and inaccurate definitions may cause misspecification. Instead, our work focuses on a more general paradigm, i.e., one-shot imitation learning, where the agent watches the human user performing the task once (Figure 1i**A**) and learns to perform the same task in unseen environments (Figure 1i**B**).

While one-shot imitation learning is a convenient paradigm for teaching new tasks to agents, it is also extremely challenging due to the fact that there could be multiple goal specifications that explain the given expert demonstration well. For instance, consider the task illustrated in Figure 1i**A**. There are multiple interpretations of the intended goal spatial relationships based on the demonstration, each of which will lead to a different task execution in a new environment (Figure 1i**C**). Without a correct understanding of the true goal, an agent cannot successfully perform the task.

To address this challenge, we proposed a novel active reward learning approach, Graph-based Equivalence Mappings (GEM), connecting offline reward learning with active reward learning. This approach improves both i) the representation of spatial goal specification and ii) the acquisition of such representation that can reveal the true spatial goal. We conducted experiments with a simulated oracle and with human subjects in a 2D physics simulation environment, Watch&Move. In each

---

*equal contribution

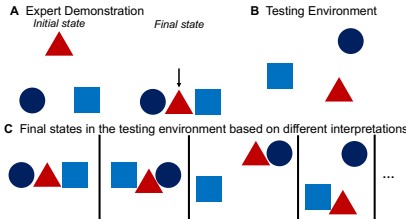

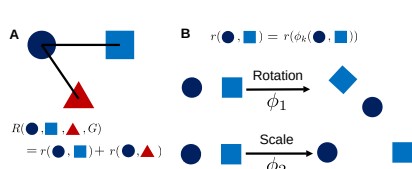

(i) We first (**A**) show a single expert demonstration for an object rearrangement task to an agent, and then (**B**) ask the agent to reach the same goal in unseen testing environments. (**C**) Multiple spatial goals can interpret the expert demonstration, each leading to a distinct task execution in the testing environments. For instance, from left to right, the four possible spatial goals shown here are i) triangle is to the right of circle and to the left of square; ii) triangle is close to circle and square; iii) triangle is to the left of circle; iv) triangle is close to square.

(ii) (**A**) We use a compositional reward function, $R$, conditioned on a graph, $G$, as the spatial goal representation. (**B**) For each edge, we may apply certain state equivalence mappings indicating a type of invariance of the intended spatial relationship between two objects, e.g. rotation-invariant ($\phi_1$), and scale-invariant ($\phi_2$).

Figure 1: Illustration of our problem setup and key elements of GEM.

task, the goal is to move the objects to satisfy spatial relationships. We compared GEM against recent baselines for imitation learning and active reward learning, and found that GEM significantly outperformed the baselines, both in terms of the generalizability and the sample efficiency (i.e., less oracle queries).

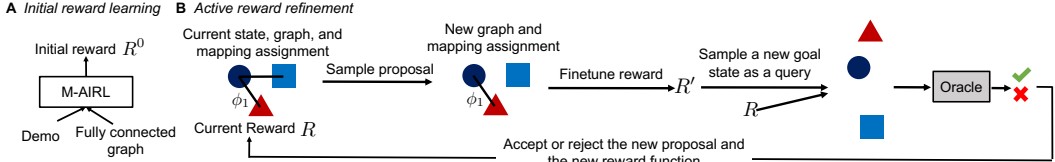

Figure 2: Overview of GEM. The reward learning consists of two phases, connecting (**A**) model-based inverse reinforcement learning that predicts an initial reward and (**B**) active reward learning. Given an expert demonstration, we first initialize the reward function conditioned on a fully connected graph ($R^0$) using model-based adversarial inverse reinforcement learning (M-AIRL). To improve generalization beyond states in the expert demonstration, we update the reward function iteratively. At each iteration we propose a new graph or a new equivalence mapping assignment for the edges in the new graph. We then finetune the reward function conditioned on the new graph using data augmented by the new equivalence mappings. To verify the fitness of the proposed graph and the equivalence mappings, we generate a new goal state that can differentiate the new reward $R'$ from the current reward $R$ as an informative query for the oracle. Based on the oracle feedback, we update the current proposal, reward function, and state accordingly. We also collect the query states as additional training data for the reward finetuning at future iterations.

In summary, our main contributions are: i) a generalizable spatial goal representation using a compositional reward function conditioned on a graph and state equivalence mappings, ii) a novel reward learning algorithm, GEM, for discovering spatial goal representations by connecting inverse reinforcement learning with sample-efficient active reward learning, and iii) a new physics simulation environment, Watch&Move, for evaluating one-shot imitation learning, focusing on generalization.

## 2 GRAPH-BASED EQUIVALENCE MAPPINGS

As a representation of a spatial goal, a reward function can adequately describe the fitness of a spatial configuration w.r.t. any intended goal spatial relationships including both logical and continuous relationships. When learned properly, the trained reward function also enables a generalization of the corresponding goal in unseen physical environments, avoiding over-imitation, unlike direct policy imitation. However, without a diverse set of demonstrations, multiple reward functions may perfectly fit the expert trajectories, and it is impossible to disambiguate the true reward solely based on the given demonstration. To solve this, we propose a novel active reward learning approach, Graph-based Equivalence Mappings (GEM) that learns a compositional reward function conditioned on a sparse graph and state equivalence mappings (Figure 1ii). As illustrated in Figure 2, GEM consists of two learning phases, combining both model-based inverse RL and active reward learning. Starting from the initial reward function that has a theoretical fitness guarantee limited to the expert demonstration, GEM iteratively refines the reward function through proposed new graphs and state equivalence mappings and verifies the new reward function via informative queries for an oracle. In this section, we first introduce our compositional reward function and then present the two-phase learning algorithm.

### 2.1 COMPOSITIONAL REWARD FUNCTION AS A SPATIAL GOAL REPRESENTATION

We represent each state as $s = (x_i)_{i \in N}$, where $N$ is a set of objects and $x_i$ is the state of object $i$. We indicate the important spatial relationships for a task by a graph, $G = (N, E)$, where each edge $(i, j) \in E$ shows that the spatial relationship between objects $i$ and $j$ is part of the goal specification. Here, we focus on pairwise spatial relationships, but it is possible to extend this to higher-order relationships. As shown in Figure 1iiA, given the graph and the state, we define a compositional reward function as a spatial goal representation to implicitly describe the goal spatial relationships for a task: $R(s, G) = \frac{1}{|E|} \sum_{(i,j) \in E} r(x_i, x_j)$ where $r(x_i, x_j)$ is the reward for edge $(i, j)$.

Spatial relationships may have certain invariance properties. For instance, relationships describing the desired distance between two objects are invariant to the rotation applied to this pair of objects. By utilizing correct invariance properties, we can transform a state seen in the training environment to a new state that has the same reward, as they represent the same spatial relationship. Essentially this process augments the data by domain randomization (Tobin et al., 2017). To model different invariance properties, we introduce a set of possible state equivalence mappings $\{\phi_k\}_{k \in K}$ as shown in Figure 1iiB, where each type of mapping $\phi_k$ can transform the states of a pair of objects $(i, j)$, and ensures that the reward component for that edge does not change, i.e., $r(x_i, x_j) = r(\phi_k(x_i, x_j))$. When applying a mapping, we may randomize the invariance aspect of the state to sample a new state. For instance, for applying the rotation-invariant mapping once, we randomize the relative orientation between the two objects for the state transformation.

We denote the mappings assignment for all the edges in a graph as $I = \{\delta_{i,j,k}\}_{(i,j) \in E, k \in K}$, where $\delta_{i,j,k}$ is a binary variable indicating whether $\phi_k$ can be applied to edge $(i, j)$. The state mapping for the whole graph is then defined as $\Phi(s, I)$, where it recursively applies mappings assigned to each edge. Note that we transform the state for each edge independently so that the transformation of one edge will not affect the state of other edges that share a common node with this edge.

In this work, we consider two types of mappings illustrated in Figure 1iiB as a gentle inductive bias provided by domain knowledge, but other types of mappings can also be applicable for different domains. Critically, we only provide a set of candidate mappings but do not assume to have the knowledge of which mappings can be applied to a specific spatial relationship, unlike Hu et al. (2020). We instead learn to assign valid mappings to each edge through active reward learning. These equivalence mappings allow us to augment the expert demonstration states and those acquired through queries, creating an infinite number of states that have equivalent spatial relationships.

### 2.2 TWO-PHASE REWARD LEARNING

Given the expert demonstration $\Gamma$, we first use M-AIRL, similar to Sun et al. (2021), to train an initial reward function $R^0(s, G^0; \theta^0)$ conditioned on a fully connected graph $G^0$, where $\theta^0$ are the parameters of the initial reward function. This initial reward function provides a good reward approximation for all the states in $\Gamma$, approaching the ground-truth with a constant offset.

To improve the generalization of the initial reward function to states beyond expert states $S_D$, we then infer the ground-truth graph $G$ and equivalence mapping assignment for the edges in the graph $I$ through active learning, and refine the reward function based on the inferred graph and the equivalence mappings. For the reward refinement, we consider three training sets, (1) all states in the expert demonstrations and their rewards based on $R^0$, $S_D = \{(s^t, r^t = R^0(s^t|G^0, \theta^0))\}_{t=0}^T$, (2) a positive state set initialized with the final state of the expert demonstration $S_+$, and (3) a negative state set, $S_-$. By querying, we collect more states for the positive set and the negative set based on the oracle judgment. We know the corresponding reward of the states in $S_D$ based on $R_0$, and assume that rewards for states in $S_+$ have higher rewards than any state in $S_-$.

The main purpose of the queries is to find equivalently good goal states that are visually different from the goal state shown in the expert demonstration. To this end, at the start of the active learning phase, we first reach the final state of the expert demonstration by applying the learned approximated value function. At each iteration $l$, we first propose a new graph $G'$ and a new equivalence mapping assignment $I'$ based on a proposal distribution $Q(G', I'|G^{l-1}, I^l)$. We describe the design of this proposal distribution in Appendix A.2. We then finetune the reward function conditioned on the proposal to obtain new parameters $\theta'$, which defines a new reward function $R(s|G'; \theta')$. To do that, we use two types of optimization. First is reward regression based on the state and reward pairs $(s, r) \in S_D$. Since we obtained equivalence states from the proposed mappings, the reward function can be optimized so that for each state $s \in S_D$, all of its equivalent states will have the same reward as $s$ itself. We formally define a regression loss as follows: $\mathcal{L}(\theta)_{\text{reg}} = \mathbb{E}_{(s,r)\sim S_D}[(R(\Phi(s, I')|G'; \theta) - r)^2]$. The second type of optimization is reward ranking. Specifically, we optimize the reward function so that the reward of a state $s_+ \in S_+$ is higher than the reward of any state $s_- \in S_-$. This gives us the second loss: $\mathcal{L}(\theta)_{\text{rank}} = \mathbb{E}_{s_+\sim S_+, s_-\sim S_-}[|(R(\Phi(s_-, I')|G'; \theta) - R(\Phi(s_+, I')|G'; \theta)|_+]$. We then combine these two loss functions to update the parameters of the reward function: $\theta' = \arg\min_\theta \mathcal{L}(\theta)_{\text{reg}} + \mathcal{L}(\theta)_{\text{rank}}$.

Based on the new reward, we generate a query to reflect the change in the hypothesis and in the corresponding reward function. The query is a goal state $s'_q$ sampled starting from the current state (i.e., $s_q^{l-1}$). Intuitively, this new state should have a high reward based on the new reward function but a low reward based on the previous reward function at iteration $l-1$. Formally, $s'_q$ is sampled by $s'_q = \arg\max_{s \in \mathcal{N}(s_q^{l-1})} R(s|G'; \theta') - \lambda R(s|G^{l-1}; \theta^{l-1})$, where $\mathcal{N}(s_q^{l-1})$ is the set of all reachable states starting from $s_q^{l-1}$ and $\lambda$ is a constant coefficient.

After reaching the new query state $s'_q$, we query for oracle feedback by asking if $s_q^l$ is an acceptable state that satisfies the goal. If acceptable, we then accept the new proposal as well as the new reward function, and augment $S_+$ with the new query $s'_q$. We also assume that $s'_q$ has a similar reward as the final demonstration state. Thus $S_D$ can also be augmented with $(s'_q, r^T)$, where $r^T$ is the reward of the final demonstration state according to the initial reward function. If the oracle feedback is negative, then we reject the new proposal and the corresponding reward function, move back to the last accepted state (i.e., $s_q^{l-1}$), and augment $S_-$ with $s'_q$. We repeat this process until reaching the maximum number of iterations $L_{\text{max}}$. Alternatively, we can also terminate the process when no sparser graphs have been accepted for a certain amount of iterations.

After the two learning phases, we apply the learned reward function to the test environments by deploying a planner based on the reward function. Since sparse graphs capture the simplest explanation of the task and are more likely to generalize, we use the most sparsest graphs accepted in the recent iterations. When there are multiple accepted graphs with the same number of edges, we use the one that has the most mappings assigned to its edges.

# 3 EXPERIMENTS

We propose a one-shot imitation learning environment, Watch&Move, where objects can be moved in a 2D physics engine. We design 9 object rearrangement tasks in the Watch&Move environment (Figure 4). Each task consists of an expert demonstration and a new testing environment. We use a reward function to measure the task completion and the displacement of all objects: $R_{\text{eval}} = \mathbb{1}_{[s^T \text{ satisfies the goal}]} - 0.02 \sum_{i=1}^N ||x_i^0 - x_i^T||_2$, where $x_i^0$ and $x_i^T$ are the first and last states of object $i$ in a testing episode respectively, and $s^T$ is the overall final state in the testing episode. We evaluate an

agent in a testing environment five times and report the average reward. Please refer to Appendix A.5 for more details about the tasks and the environment.

We compare GEM with M-AIRL and with ReQueST (Reddy et al., 2020), a recent active reward learning approach that estimates a reward that can generalize to environments with different initial state distributions. For the ablated study, we also evaluate variants of GEM, including i) GEM without minimizing the previous reward for the query generation, ii) GEM with a fully connected graph (no new graph proposals), iii) GEM without applying any equivalence mappings, and iv) GEM trained with randomly generated queries. During testing, given the reward function learned by each method, we sample a goal state by maximizing the reward. We then evaluate sampled goal states based using our reward metric defined above. Finally, we provide an oracle performance based on the optimal goal states generated by the oracle in the testing environments.

We first conducted an evaluation on 8 Watch&Move tasks with a simulated oracle that gives feedback for a query based on whether the state in the query satisfies the goal definition. We report the reward obtained in the testing environment using models trained with different methods with different numbers of queries in Figure 3i (see full results in Figure 5). The results show that the initial reward trained by M-AIRL failed to reach the goal in the tesing environment. However, with the active reward refinement enabled by GEM, the reward function was greatly improved, significantly outperforming all baselines.

To evaluate how well GEM can work with real human oracles, we conducted a human experiment on Task 9, where we recruited three human participants as oracles. The participants gave their consent and the study was approved by an institutional review board. We plot the rewards in the testing environment in Figure 3ii, which demonstrates that the reward function trained by GEM reaches a good performance in the new environment within 30 iterations, significantly outperforming ReQueST.

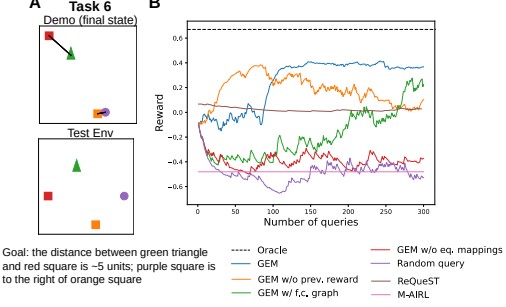

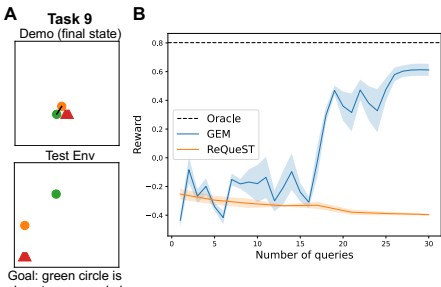

(i) (**A**) Illustration of Task 6 used in the simulated oracle experiment. (**B**) Testing performance of different methods trained with a simulated oracle on Watch&Move Task 6. We ran each method using three random seeds and show the standard errors as the shaded areas.

(ii) (**A**) Illustration of Task 9 used in the human experiment. (**B**) Testing performance of GEM and ReQueST with three human oracles for Task 9. The shaded areas indicate the standard errors.

Figure 3: Simulated oracle and human oracle on Tasks 6 and 9.

## 4 CONCLUSION

We have proposed a reward learning algorithm, GEM, for performing one-shot generalization for an object rearrangement task. The experimental results show that GEM was able to discover meaningful spatial goal representations. It significantly outperformed baselines, achieving a better generalization in unseen testing environments as well as a greater sample efficiency. We believe that GEM is a step towards solving the extremely challenging problem of one-shot imitation learning.

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

## A  IMPLEMENTATION DETAILS

### A.1  STATE EQUIVALENCE MAPPINGS

**Applying a mapping to an edge once**. When applying a mapping to an edge, we randomly sample a variable necessary for the mapping. For the the rotation-invariant mapping, $\phi_1(x_1, x_2)$, we sample a random angle $d \sim \text{Uniform}(-\pi, \pi)$, and rotate $i$ around $j$ by the angle of $d$. For the scale-invariant mapping $\phi_2$, we randomly sample a scale $\rho \sim \text{Uniform}(0.1, 10.0)$, and move $i$ so that its distance from $j$ changes by the scale of $\rho$ while the relative orientation from $i$ to $j$ remains the same.

**State transformation given all mappings**. For each graph, there is a corresponding mapping assignment for all edges $I = \{\delta_{ijk}\}_{(i,j)\in E, k\in K}$. We apply a mapping $\phi_k$ to edge $(i,j)$ when and only when $\delta_{ijk} = 1$. If multiple mappings are assigned to an edge, they will be applied recursively. E.g., if $\phi_1$ and $\phi_2$ are assigned to $(i,j)$, the final state transformation for this edge is $\phi_2(\phi_1(x_i, x_j))$. Let $\tilde{x}_{ij}$ be the transformed edge, then the final transformed state is $\Phi(s, I) = \{\tilde{x}_{ij}\}_{(i,j)\in E}$, and the corresponding reward becomes:

$$R(\Phi(s, I)|G) = \frac{1}{|E|} \sum_{(i,j)\in E} r(\tilde{x}_{ij}). \tag{1}$$

Note that since the state transformations are applied to each edge independently, any change in one edge will not affect other edges. During reward finetuning, we apply state transformation based on the proposed mapping assignment to each batch, so that the trained reward function reflects the intended invariance represented in the assigned equivalence mappings for each edge.

### A.2 PROPOSAL SAMPLING

Each proposal consists of a new graph and a new equivalence mapping assignment. Therefore, there are two general types of proposal sampling – (1) graph sampling and (2) equivalence mapping assignment sampling. We use a prior probability $q_{\text{type}}$ to decide the type of sampling for each iteration. At each iteration, we first sample $u \sim \text{Uniform}(0, 1)$. If $u < q_{\text{type}}$, we choose to sample a new equivalence mapping assignment; otherwise, we sample a new graph.

To sample a new graph, we can either add an edge or remove an edge. We define the chance of removing an edge as $q_{\text{remove}}$. Then we sample $u \sim \text{Uniform}(0, 1)$. When $u < q_{\text{remove}}$, we sample one of the edges from $G^{l-1}$ to remove; otherwise, we randomly add an edge that does not exist in $G^{l-1}$. Note that we consider undirected graphs in this work; we also avoid removing all edges to ensure a valid graph-based reward function.

To sample a new equivalence mapping assignment, we uniformly sample an edge $(i, j) \in E$ and a type of mapping $k \in K$, and change the corresponding assignment, i.e., $\delta'_{ijk} = 1 - \delta^{l-1}_{ijk}$.

### A.3 NETWORK ARCHITECTURE

The discriminator reward and value networks are implemented by a graph-based architecture, as opposed to the MLP architecture used in the original AIRL version. The input is the observation representation $s$ and graph $G$. We construct edge representations by concatenating every pair of object representations in the observation. We then apply 4 x (fully connected layers + ReLU) to each edge representation. We apply a final fully connected layer to each edge embedding to output a single value for each edge. The final reward is the average edge value **only** for edges in $G$. This specific choice of architecture is what ties the reward function to the specific graph hypothesis, by defining an edge mask input to the reward.

### A.4 TRAINING DETAILS AND HYPERPARAMETERS

**Model-based state-based AIRL**: We build upon an AIRL implementation (Wang et al., 2020) and add a model-based generator. For each task, M-AIRL is executed for 500k generator steps, the expert batch size is the length of the expert demonstrations. The discriminator is updated for 4 steps after every model-based generator execution. The model based generator samples approximately 2k steps on each iteration.

**Query reward finetuning**: We apply 10k network updates (5k updates for human experiments) per query iteration. For optimizing the network, we use Adam optimizer (Kingma & Ba, 2014) with a learning rate of 0.0003. For each update, we sample a batch of 16 states for the regression loss and a batch of 16 pairs of positive and negative states for the reward ranking loss.

**Query selection**: we set $\lambda = 0.2$ in $s'_q$ subsection 2.2.

**Random query variant**: we always assume a fully connected graph and do not assign equivalence mappings to the edges. We collect the positive and negative sets and refine the reward function using the same loss function defined in subsection 2.2.

### A.5 WATCH&MOVE ENVIRONMENT

Figure 4 illustrates the setup (the expert demonstration, the test environment, and the goal description) for each Watch&Move task. The demonstrations of these tasks present various sources of confusion. For instance, there can be irrelevant objects (e.g.,the purple trapezoid and the red circle in Task 8) or goal objects that are never moved in the demonstration due to their initial states being satisfactory (e.g., the blue trapezoid and the blue rectangle in Task 3). Expert demonstrations were

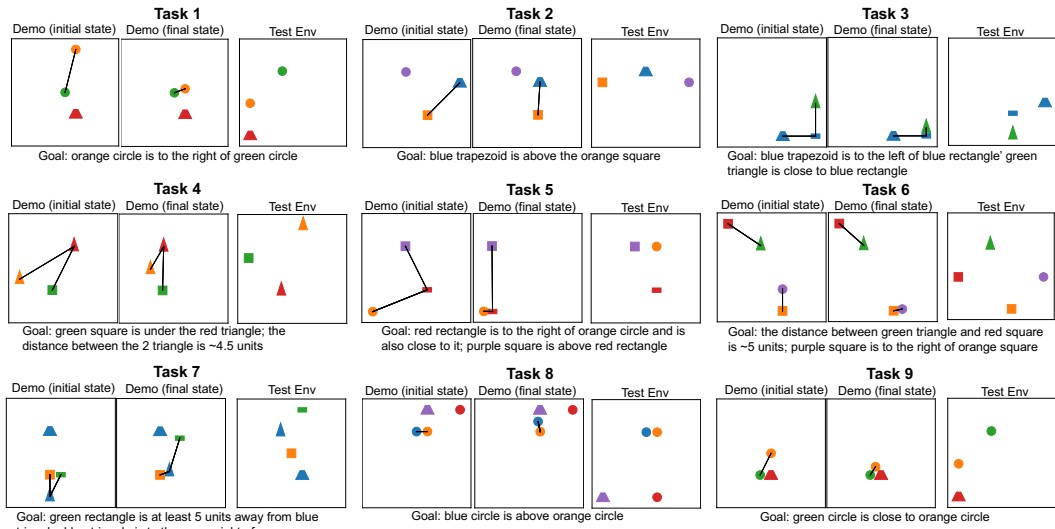

Figure 4: Illustration of all 9 Watch&Move tasks. For each task, we show the demonstration, the testing environment, and the goal definition. We also show the goal relationships by visualizing the corresponding edges.

Table 1: Definition of goal relationships in Watch&Move tasks. For reference, the radius of a circle is 0.8 units.

| PAIRWISE GOAL RELATIONSHIP | DEFINITION |
|---|---|
| CLOSE [(SRIVASTAVA ET AL., 2022)] | DISTANCE BETWEEN OBJECTS IS LESS THAN 2.5 UNITS |
| LEFT/RIGHT [(JOHNSON ET AL., 2017; YAN ET AL., 2020)] | ANGLE BETWEEN OBJECTS AND X AXIS IS LESS THAN $\frac{\pi}{10}$ |
| ABOVE/BELOW [(YAN ET AL., 2020)] | ANGLE BETWEEN OBJECTS AND Y AXIS IS LESS THAN $\frac{6\pi}{10}$ |
| DIAGONAL | BOTH COORDINATES OF ONE OBJECT ARE LARGER THAN THE OTHER'S |
| DISTANCE X | THE DISTANCE BETWEEN OBJECTS IS WITHIN A 1.5 UNIT BUFFER AROUND X |
| AT LEAST WITHIN DISTANCE X | THE DISTANCE BETWEEN OBJECTS IS GREATER THAN X |

created with a planner introduced in PHASE (Netanyahu et al., 2021), with a length ranging from 8 to 35 steps.

The state space in Watch&Move is represented by $s \in \mathbb{R}^{N \times 13}$ where $N$ is the number of objects in the environment. 13 dimensions are composed of the coordinate of the object center, the object's angle, and one hot encodings of the object shape and color. The action space is discrete, containing 11 possible actions per object (8 directions, turning right and left and stopping), and the object id. The action space is proportional to the number of objects. We use PyBox2D[1] to simulate the physical dynamics in the environment.

The goal relationships used to create Watch&Move are specified in Table 1. These could be easily extended to any pairwise spatial relation, such as touching, covering, no contact, orientation, etc.

## A.6 REQUEST

ReQueST (Reddy et al., 2020) is a method for estimating a reward ensemble that can safely generalize to environments with different initial state distributions. ReQueST generates queries from a generative model that optimizes four objectives. Each state in each query receives accepted or

---

[1]https://github.com/pybox2d/pybox2d

rejected feedback from an oracle and is used as a positive or negative example in reward training. To ensure a fair comparison with our approach, we implement ReQueST with the following changes.

- The original method does not use an expert demonstration, therefore we pre-train the ensemble reward functions with the expert demonstration.

- The reward architecture is similar to ours, where the final edge embeddings are sum-pooled and fed to a fully connected layer followed by softmax for the classification. Note that here we only have two classes – neutral and good.

- The original paper learns a world model from random sampling in the environment. We use the ground truth world model provided by the physics simulation, similar to GEM.

- Each query is of length 1 (as in the pointmass environment in ReQueST) sampled similarly to the AIRL generator sampling in GEM. Starting from the final state of the expert demonstrations, each query is sampled relative to the last sampled point, matching with the query generation procedure in GEM.

We use an ensemble of 4 rewards (as in the pointmass environment in ReQueST),and train them using the Adam optimizer with a learning rate of 0.0003. We have 1k pre-training steps and 10 training steps per every 4 queries.

## B  ADDITIONAL RESULTS

### B.1  GEM RESULTS

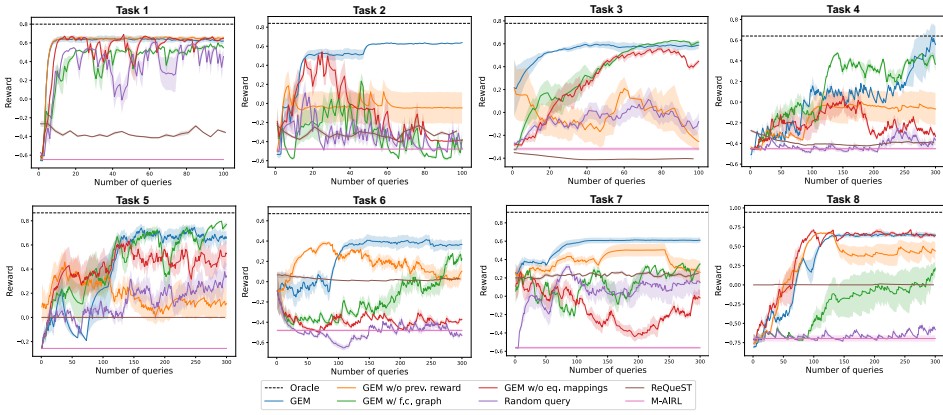

Figure 5: Testing performance of different methods trained with a simulated oracle on 8 Watch&Move tasks (Task 1 to 8). We plot the reward metric in the testing environment using the learned model as a function of the number of queries. Note here the M-AIRL baseline provides the initial rewards for GEM and all of its variants and is not updated with the queries. We ran each method using three random seeds and show the standard errors as the shaded areas.

### B.2  INFERRED GRAPHS

We show the inferred graphs and the equivalence mapping assignments for all 9 tasks in Figure 6. The inferred graphs correctly identified the goal relationships. The assigned equivalence mappings also revealed the invariance properties of the intended spatial relationships in all but two cases. The two exceptions are i) the edge between the red square and the green triangle in Task 6 and ii) the edge connecting the blue circle and the orange circle in Task 8. In general, when the inferred graph indicates only the necessary relationships and the assigned equivalence mappings correspond to the correct types of invariance, the trained reward will have a stronger generalization in the testing environment.

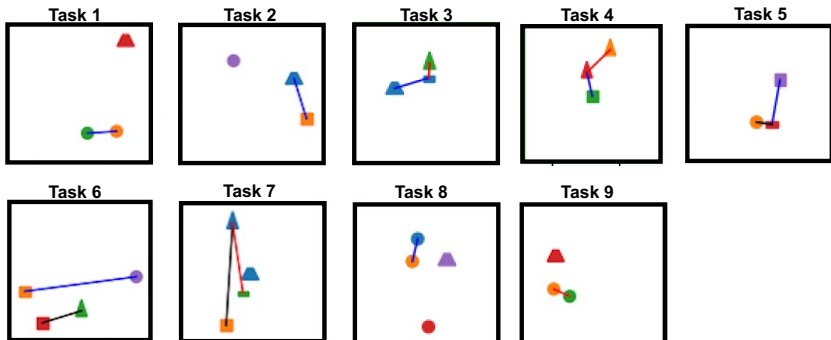

Figure 6: Illustration of the inferred graphs and equivalence mapping assignments as well as the corresponding query states that lead the proposal acceptance for all 9 tasks (from one of the three runs). The colors of the edges indicate the assigned mappings. Black: no mapping is assigned; red: the rotation-invariant mapping is assigned; blue: the scale-invariant mapping is assigned; purple: both the rotation-invariant and the scale invariant mappings are assigned.

## B.3 QUERY VISUALIZATION

Figure 7 depicts typical queries that are sampled by GEM to verify the proposed graphs and the equivalence mapping assignment. The examples here demonstrate how the informative queries were able to help differentiate two different reward functions.

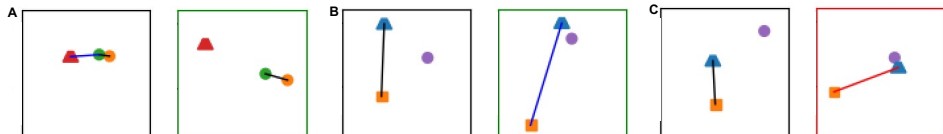

Figure 7: (**A**)-(**C**) are example queries showing the effect of the new graph and equivalence mapping assignment. Each example first shows the state and proposal before the query, and then shows the new proposal and the sampled query state. The colors of the edges indicate the assigned mappings. Black: no mapping is assigned; red: the rotation-invariant mapping is assigned; blue: the scale-invariant mapping is assigned; purple: both the rotation-invariant and the scale-invariant mappings are assigned. The colors of the boxes indicate oracle feedback (red: reject, green: accept). In (**A**), an edge was removed, and the irrelevant object was consequently placed far away from the remaining objects. The example in (**B**) shows that when the scale-invariant mapping was assigned to an edge, the sampled query changed the distance between the objects connected by that edge while generally preserving the relative orientation between the two. On the other hand, when the rotation-invariant mapping was assigned as in (**C**), one of the objects was rotated around the other connected object and there was little change in the distance between the two objects.

