# OpenReview forum: "Discovering Generalizable Spatial Goal Representations via Graph-based Active Reward Learning"
_ICLR.cc/2022/Workshop/OSC — ICLR2022 OSC  Poster_

### Official Review · Reviewer_3zbj · 2022-03-12
**New reward learning method for one-shot imitation learning**

**Rating:** 2
**Confidence:** 2

**Review:**

This paper proposes a new reward learning approach (“GEM”) for the challenging task of one-shot imitation learning on toy-ish object rearrangement tasks. Given an expert demonstration, in a first step, GEM is learning a compositional reward function assuming a fully connected graph between all objects in the scene. In a second step, this reward function is being iteratively fine-tuned by querying the expert oracle for various new graph mappings. The authors test their method on a newly proposed 2D environment for evaluating generalization capabilities in one-shot imitation learning settings.

Pros
- One-shot imitation learning for object rearrangement is a very challenging task and the authors do a good job at explaining the studied problem setting
- As far as I can tell, the proposed method is clearly novel and original and relevant design decisions are well motivated
- Extensive experiments on the newly proposed environment show that the method is outperforming various baselines
- Many ablations disentangle the individual contributions of the different parts of the method.
- The experiments with the human oracles are pretty cool!

Cons
- Section 2 is rather difficult to follow. Maybe clarity could be improved here.
- Given the dataset is being described as a stand-alone contribution, I think it would help if this dataset is being described in a bit more detail in the main paper.
- I didn’t follow why the method is supposed to select the sparsest graph in the end. I don’t understand why this is a reasonable assumption unless it is known that sparsely connected graphs are a feature of these particular environments.
- What is the reason for the second term in R_eval?

---

### Official Review · Reviewer_wiVk · 2022-03-14

**Rating:** 2
**Confidence:** 2

**Review:**

Summary: The paper proposes an object-centric architecture for reward functions that can be quickly learned using a single expert demonstration followed by a small number of queries asked to an oracle to find out whether the task was successful or not. The reward function consists of a reward function, a graph, and equivalence function assignments for each graph edge. The reward function is compositional and object-centric in the sense that the full reward is an average of reward computed per object pair. The graph determines which object pairs should be considered during reward calculation based on which edges are present in the graph. Finally, to each edge, a set of equivalence functions are assigned from a predefined function library. When the contribution of each object pair to the reward is computed, the object pair can be fed to its equivalence functions to nullify the effects of transformations (such as rotation or scaling) that do not violate the task goal.

The learning takes an iterative active-learning-based approach. An initial reward function is first learned using M-AIRL which returns a reward function and a fully connected graph. In each iteration, a slightly different graph and equivalence assignment proposals are sampled with respect to the previous ones and then a reward function is trained. A final state is generated based on the newly trained reward function and the final state is queried against the oracle to find out if this state is acceptable. Based on this feedback the proposal (along with the reward function) is either accepted or discarded and the iterations continue until a stopping criterion.

Pros: Demonstrates an interesting use-case for object-centric representations that shows sample efficient learning of reward functions and generalization to unseen environments.

Cons: Adding a brief description of M-AIRL could be helpful.

Conclusion: The paper is interesting and relevant for the workshop.

---

### Decision · Program_Chairs · 2022-03-21

**Decision:**

Accept (Poster)

**Comment:**

The reviewers agree the paper should be accepted at the workshop. Congratulations!